# Flexural Strength and Hardness of Filler-Reinforced PMMA Targeted for Denture Base Application

**DOI:** 10.3390/ma14102659

**Published:** 2021-05-19

**Authors:** Abdulaziz Alhotan, Julian Yates, Saleh Zidan, Julfikar Haider, Nikolaos Silikas

**Affiliations:** 1Division of Dentistry, School of Medical Sciences, University of Manchester, Manchester M13 9PL, UK; julian.yates@manchester.ac.uk (J.Y.); saleh_0072002@yahoo.co.uk (S.Z.); j.haider@mmu.ac.uk (J.H.); nikolaos.silikas@manchester.ac.uk (N.S.); 2Dental Health Department, College of Applied Medical Sciences, King Saud University, Riyadh 11454, Saudi Arabia; 3Department of Dental Materials, Faculty of Dentistry, Sebha University, Sebha, Libya; 4Department of Engineering, Manchester Metropolitan University, Manchester M1 5GD, UK

**Keywords:** PMMA, ZrO_2_ nanoparticle, TiO_2_ nanoparticle, E-glass fibre, flexural strength, surface hardness

## Abstract

The aim of this work was to evaluate the flexural strength and surface hardness of heat-cured Polymethyl methacrylate (PMMA) modified by the addition of ZrO_2_ nanoparticles, TiO_2_ nanoparticles, and E-glass fibre at different wt.% concentrations. Specimens were fabricated and separated into four groups (n = 10) to measure both flexural strength and surface hardness. Group C was the control group. The specimens in the remaining three groups differed according to the ratio of filler to weight of PMMA resin (1.5%, 3%, 5%, and 7%). A three-point bending test was performed to determine the flexural strength, while the surface hardness was measured using the Vickers hardness. Scanning Electron Microscope (SEM) was employed to observe the fractured surface of the specimens. The flexural strength was significantly improved in the groups filled with 3 wt.% ZrO_2_ and 5 and 7 wt.% E-glass fibre in comparison to Group C. All the groups displayed a significantly higher surface hardness than Group C, with the exception of the 1.5% TiO_2_ and 1.5% ZrO_2_ groups. The optimal filler concentrations to enhance the flexural strength of PMMA resin were between 3–5% ZrO_2_, 1.5% TiO_2_, and 3–7% E-glass fibre. Furthermore, for all composites, a filler concentration of 3 wt.% and above would significantly improve hardness.

## 1. Introduction

Over the last few decades, research on dental biomaterials has significantly progressed, leading to notable improvements in the associated material properties and technologies and fundamentally transforming dental materials and restorative practices [1,2]. The major challenges that dental clinicians traditionally encounter in prosthetics and restorations pertain to the lack of biocompatibility of materials, the difficulty in achieving a natural appearance, and the inability to develop a material that can withstand exposure to the harsh oral environment [1,3,4]. PMMA has been a preferred choice in the production of denture bases since its introduction in 1930s [5,6]. While dental implants are increasingly being used as substitutes for natural teeth in both partially dentate and edentulous patients, PMMA remains the preferred choice for denture base construction [7]. The popularity of PMMA as a polymeric material among those currently available can be attributed to its functional properties [8,9]. PMMA has achieved its popularity due to several factors; including, easy manipulation and processing, fabrication with affordable equipment, aesthetically appealing, and biocompatibility [5,8,10,11,12]. However, as conventional PMMA denture bases are relatively brittle and weak, they have a propensity for mechanical failure, leading to a high risk of fracture [5,11,13]. Several clinical reports have concluded that complete maxillary dentures are at risk of midline fracture [14,15]. The majority of fractures result in the mouth during normal function [2]. This phenomenon can be attributed to the resin fatigue that results from deep scratches and stress intensification (mastication) [11,14]. The ever-changing dental requirements entail that denture materials are required to exhibit optimal mechanical performance [5,16]. Therefore, many investigators have attempted to modify PMMA denture bases in a bid to improve its mechanical performance [5,11,13,17]. Researchers have consistently focused on the use of filler materials as a means of modifying the properties of polymer composites and blends [10,12,18,19,20,21]. Numerous studies have investigated how various filler materials, such as metal-oxide nanoparticles (ZrO_2_, TiO_2_, Al_2_O_3_) [5,21,22], polymeric fibre (nylon, carbon, aramid, polyethylene) [14,23], and glass fibre [24,25] can improve the mechanical properties of PMMA denture base. However, while the findings of the existing studies have sometimes been promising, they have also frequently contradicted each other [26].

During the mid-1990s, researchers started investigating the use of novel fibres as potential fillers for enhancing the mechanical properties of PMMA denture bases [27]. These novel fibres were promising as a means of reinforcing denture bases due to their strength, biocompatibility and aesthetics [10,23,28]. However, the performance of polymer composites depends on many factors, such as the nature of the matrix bonding, resins used, manufacturing conditions, and fibre type, percentage, length, diameter, or orientation [9,10,17]. Amongst the fibres used in this application, E-glass fibre has a high level of chemical resistance, is comparatively cost effective, achieves an excellent aesthetic appearance and possesses good biological properties [29].

Recently, research has focused predominantly on nanoparticles, which are widely recognized as they are offering advantageous characteristics due to their size, shape, composition, and ability to enhance the existing properties of polymers [26,30]. The emerging science and technology in the field of nanofillers are promising for the fabrication of PMMA polymer composites [15,18,19,20,21]. A relatively new material called zirconium oxide (ZrO_2_)—also referred to as Zirconia—has grown in popularity [31,32] as it is a typical bioceramic and, therefore, exhibits a good bioactivity and biocompatibility. Zirconia also offers exceptional high flexural strength, fracture toughness and durability, in addition to good aesthetics [31,32,33]. Another nanomaterial that has attracted significant interest is titanium oxide (TiO_2_). TiO_2_ offers suitable cost and mechanical properties, is chemically stable and is non-toxic [34].

The materials that are commonly used in dental prostheses are associated with a risk of deformation and related fracture [1,5]. Flexural loading simulates clinical situations in which dentures are exposed to complex of forces during service; for example, compressive, tensile, and shear forces [14,16]. Conventional PMMA is a brittle material; therefore, it often fails during complex loading [1]. Prior fracture, scratches, or small flaws on the surface of PMMA dentures can generate crack propagation, leading to the eventual failure of the denture [4,16]. Thus, softer materials tend to develop scratches during the acts of daily living, such as brushing, chewing, or cleaning. These scratches can ultimately lead to fracture [35,36].

Many studies have concluded that the use of various fillers to reinforce heat-cured PMMA denture base resins can enhance their mechanical properties [18,20,26,30]. However, the existing literature does not include any systematic studies on the impact nano-TiO_2_, glass fibre, and nano-ZrO_2_ fillers have on the hardness and flexural strength of heat-cured PMMA denture base resins. Thus, further studies are required to pinpoint the optimal quantity of nano-TiO_2_, glass fibre, and nano-ZrO_2_ required to enhance the longevity and performance of PMMA denture base resins. The current study explores the influence of ZrO_2_ and TiO_2_ nanoparticles on the mechanical properties of PMMA acrylic resin, along with the popular glass fibre type “E”. E-glass fibre is an accepted reinforcement for PMMA material [6,29,37]. The optimum concentration for dental reinforcements needs to be studied. Zirconium, titanium nanoparticles, and E glass fibres were systematically varied at concentrations of (1.5 wt.%, 3 wt.%, 5 wt.%, and 7 wt.%). The research was underpinned by the null hypothesis that the inclusion of filler would not improve the surface hardness and flexural strength of the specimens.

## 2. Material and Methods

### 2.1. Materials

Table 1 presents the materials employed in this study. Conventional heat-polymerized acrylic resin consists of polymethylmethacrylate (PMMA) powder and liquid methyl methacrylate (MMA) monomer. In addition to the use of commercial ZrO_2_, TiO_2_ nanoparticles, and chopped E-glass fibre as filler materials, silane coupling agent (3-Trimethoxysilyl propyl methacrylate) and ethanol were employed to treat the surfaces of the filler before adding to PMMA.

### 2.2. Method

#### 2.2.1. Specimen Preparation

In total, 130 rectangular specimens (64 mm in length, 3.3 ± 0.2 mm in thickness, and 10 ± 0.2 mm in width) were fabricated. The samples were divided into four different groups: one control group (pure, heat-cured PMMA) and three experimental groups that differed according to the reinforcement materials incorporated into the PMMA: ZrO_2_, TiO_2_, and E-glass fibres. Each of the three reinforced PMMA groups (n = 10 each group) were further sub-divided into four subgroups according to the concentration of filler used: 1.5%, 3%, 5%, and 7%) (Table 2). All the specimens were prepared according to the manufacturer’s instructions and tested per ISO specifications [38,39], according to previous studies [5,6,18,40].

#### 2.2.2. Selecting Appropriate Filler/Saline Percentages

Pilot studies determined that the percentages of the silanized fillers used should be 1.5 wt.%, 3 wt.%, 5 wt.%, and 7 wt.%. Table 3 illustrates the composition of all the specimen routes employed by this study. A ratio of heat-cured acrylic resin powder and monomer (P/L) ratio of 21 g:10 mL was employed in accordance with the recommendations of the manufacturer; scaled appropriately, this meant that a proportion of 12 g to 5.7 mL for the mould was employed.

To select the silane, a pilot study was performed using different quantities of saline monomer (y-MPS), at 3 wt.%, 5 wt.%, and 10 wt.% Taking into account the results of this study and those of previous experiments, a 3 wt.% percentage weight of (y-MPS) was selected.

#### 2.2.3. Treating Surface of Fillers (Silanization)

The surfaces of the fillers were modified using a 3 wt.% of silane coupling agent (γ-MPS) before mixing with PMMA to promote improved chemical bonding between the fillers and the acrylic resin matrix [10,18]. A speed mixer (DAC 150.1 FVZK, High Wycombe, Buckinghamshire, UK) was used to mix 15 g of each filler with 70 mL of ethanol in a 100 mL plastic container for 10 min (1500 rpm), ensuring that the surfaces of the filler were adequately cleaned and coated evenly with ethanol solution. Then, 0.45 g (3 wt.%) of γ-MPS was added to the resultant suspension, and a magnetic stirrer (200 rpm over 2 h at room temperature) was employed to achieve a uniform consistency. The suspension was then refluxed for 4 h at 50 °C. When the reaction was complete, the resulting mixture was cooled, split into two equal parts, and placed into 50-mL plastic tubes sealed with a plastic lid. These tubes were then placed in a centrifuge (Heraeus, UK) and rotated for 20 min at 4500 rpm and a temperature of 23 °C. The clear supernatant (containing both unreacted and condensate γ-MPS) was then decanted, leaving the sediment behind. The sediment comprised filler silanized with γ-MPS; this element comprises the centrifuged sample. The plastic tubes were then covered with perforated aluminium foil and placed into the Genevac machine (Genevac EZ-2 series, SP Scientific Company, UK) for drying for 3 h at 50 °C, thus allowing solvent evaporation. This process produced the silanized fillers.

#### 2.2.4. Combining Nanoparticles and E-Glass Fibre with PMMA/MMA

An electronic balance (Ohaus Analytical, Parsippany, NJ, USA) was employed to weigh conventional heat-polymerized powder and different quantities of silanized fillers. The silanized ZrO_2_ nanoparticles were placed in a speed mixer with the MMA monomer for 10 min at 1500 rpm; this allowed a homogenous suspension to form and avoided aggregation of ZrO_2_. The resultant product was regarded as the modified monomer. For each group, the modified monomer was mixed with PMMA powder in a ratio of 21 g powder and 10 mL modified monomer as per manufacturer’s guidelines. As such, the ratio of mixing acrylic resin/filler to monomer was 12 g powder/filler to 5.7 mL monomer in each mould. This process was then repeated with the TiO_2_ nanoparticles.

To mix PMMA/MMA with silanized E-glass fibres, an identical powder/monomer ratio of 12 g to 5.7 mL was also employed. The necessary quantity of chopped E-glass fibres were dampened using 4.3 mL of MMA. These fibres, which were bundled together, were spread out and carefully manipulated by hand to allow for thorough saturation. Subsequently, 0.400 g of the PMMA powder was added to the liquid and mixed for 10 s. This was done six times to make sure that the materials blended properly. The mixture was then stirred for 2 min longer to make sure that the chopped fibres adequately embedded in the MMA matrix. It could then be seen that the fibres were well distributed throughout the liquid. Finally, the remaining resin powder required (9.6 g) was added to this mixture of E-glass fibre and PMMA, and then the last of the monomer (MMA) liquid (1.4 mL) was added.

With both mixing methods, the mixture of PMMA/MMA and filler was stirred with a spatula for approximately a minute to make sure that the monomer moistened all powder and filler. An appropriate container was used that ensured the monomer could not evaporate. After around 20 min at room temperature, a packing (dough-like) consistency resulted. Having reached this stage, the resultant dough was manipulated by hand into the mould that was painted with a thin coating of separating medium to prevent the dough from attaching to the mould during the polymerization process. The mould was then closed and placed in a hydraulic press (Sirio Dental, Meldola (FC), Italy); the force was gradually increased to 10.34 MPa. Then, the mould was placed in a clamp that ensured a tight seal for final closure. The mould and clamp were then placed under water in a curing unit (Wassermann Dental-Maschinen GmbH, Hamburg, Germany), starting gradually from room temperature to 74 °C for 90 min, followed by 30 min at 95 °C. On completion of the polymerization cycle, the mould was allowed to cool for 30 min at room temperature before opening, to reduce the chances of the specimens becoming stressed or warped. Finally, the specimens were taken out of the mould and abraded with series of silicon carbide papers (Buehler Ltd. Esslingen, Germany). Then the surfaces of the specimens were polished and smoothened using a lapping machine (MetaServ 250, Buehler Ltd. Esslingen, Germany).

### 2.3. Mechanical Measurments Procedures

#### 2.3.1. Flexural Strength Measurement

Ten specimens from each group were stored separately in distilled water in an incubator at 37 °C for 7 days before testing in line with ISO 1567 standard [38]. The flexural strength of the specimens was then measured using a three-point bending test performed in a universal testing machine (INSTRON 3344.Software: Bluehill 3, Norwood, MA, USA) according to the ISO 1567 standard [38]. Upon removal from the distilled water, each specimen was immediately placed at a parallel angle on two supports. The distance between the two supports was kept at 50 mm. At room temperature, a 500 N load cell was applied vertically at the specimen midpoint at a crosshead speed of 5 mm/min until fracture occurred. The equation shown below (Equation (1)) was used to calculate the flexural strength ($\sigma_{f}$) of each specimen in MPa:(1)$$\sigma_{f}=\frac{3\mathrm{FL}}{2{\mathrm{bh}}^{2}}$$
where, F is the peak load applied to the specimen in Newtons (N), L is the support span length (50 mm), b is the specimen width in mm, and h is the specimen thickness in mm.

#### 2.3.2. Surface Hardness Measurement

Hardness was assessed with a digital microhardness tester (FM-700, Future Tech Corp., Kawasaki, Japan). An indenter point in the form of a square-based pyramid was applied at a load of 300 g for 15 s at room temperature after 7 days of storage at 37 °C. Five indentations were made at different points along each specimen on the same surface side, with a minimum distance of 1 mm between any two indentations. The mean hardness value of each specimen group was then calculated. The Vickers microhardness (HVN) value was calculated using Equation (2).
(2)$$\mathrm{HV}=1.8544\;\frac{F}{d^{2}}\;$$
where, F is the applied load in kilogram-force (kgf), and d is the arithmetic mean of the two diagonals, d_1_ and d_2_ from surface area of the indentation in mm.

### 2.4. Fracture Surface Analysis

A scanning electron microscope SEM (Carl Zeiss Ltd., 40 VP, Smart SEM, Cambridge, UK) was employed to study the fractured surface of the control group specimen. Specimens at 7 wt.% concentrations from each subgroup were selected at the point of loading after the flexural strength test to assess the adhesion of fillers with the resin matrix, possible porosity and other defects. The specimens were mounted onto aluminium stubs and sputter-coated with gold before a secondary electron detector at an acceleration voltage of 2.0 kV was used to perform the SEM visualization at different magnifications. The shape and size distribution of the pure PMMA powder and fillers alone were also examined.

### 2.5. Statistical Analysis

The flexural strength and surface hardness data were statistically analysed using a statistical software (SPSS statistics version 25, IBM, New York, NY, USA). According to the findings of the Levene and Shapiro–Wilk tests, the p-values were not statistically significant, indicating that the data were normally distributed with homogeneous variance. A one-way analysis of variance (ANOVA) was performed at a *p* ≤ 0.05 significance level with a Tukey post-hoc test.

## 3. Results

### 3.1. Physical Characteristics of Particles

As shown in Figure 1A, according to the outputs of the SEM analysis, the particle size of the PMMA powder ranged from 10 µm to 90 µm, with a mean size around 50 µm. The average size of the ZrO_2_ nanoparticles ranged between 25 nm and 50 nm for the individual particles and 150 nm to 300 nm for the clusters, as can be observed in Figure 1B. The average size of the TiO_2_ nanoparticles ranged from 20 nm to 30 nm for the individual particles and 100 nm to 300 nm for the clusters, as can be seen in Figure 1C. The length and diameter of the E-glass fibres were between 2.5–3.2 mm and 14–18 μm respectively, as shown in Figure 1D.

### 3.2. Flexural Strength

Figure 2 and Table 4 present the means and standard deviations of flexural strength across the tested groups. The outputs of the ANOVA revealed that there was a statistically significant difference in the flexural strength of the specimens in the E-glass fibre and the ZrO_2_ nanoparticle groups when compared to the specimens in Group C (*p* < 0.05). However, there was no statistically significant difference (*p* > 0.05) between the flexural strength of the specimens reinforced with TiO_2_ nanoparticles groups and those in Group C. The results of the Tukey test revealed that the significant differences (*p* < 0.05) in flexural strength was observed in the specimen groups Z3, E5, and E7 (98.4 ± 8.3 MPa, 101.2 ± 10.4 MPa and 105 ± 10.6 MPa) compared to those in Group C (89.2 ± 6.3 MPa). The highest flexural strength value was observed in Group E7 (105 ± 10.6 MPa), while Group T7 (83.5 ± 7.2 MPa) had the lowest flexural strength among the reinforced groups.

Of the groups that were reinforced with ZrO_2_ nanoparticles, there was a significant difference (*p* < 0.05) in flexural strength between Group Z3 and Group Z7. Overall, the mean flexural strength in Groups Z1.5 and Z3 increased to 92.6 and 98.4 MPa respectively. However, the mean flexural strength of the specimens in the Z5 group slightly dropped to 95.8MPa, while that of Group Z7 dropped sharply to 88.3MPa and exhibited a lower flexural strength than that in Group C. However, this drop in strength values was not statistically significant (*p* > 0.05).

As the concentration at various ratios 1.5 wt.%, 3 wt.%, 5 wt.%, and 7 wt.% of filler increased, the flexural strength of the specimens reinforced with TiO_2_ nanoparticles gradually reduced from 91.5 ± 8.3, 88.1 ± 11.5, 86.3 ± 9.2 to 83.5 ± 7.2 MPa respectively. However, no statistically significant (*p* > 0.05) difference was found between the specimens in these groups. A negative correlation (R^2^ = 0.9233) was found between the strength and the TiO_2_ particle concentrations.

The flexural strength of the groups modified with E-glass fibre was positively correlated (R^2^ = 0.9358) with the E-glass concentrations, i.e., E1.5 (94.1 ± 6.9 MPa) had the lowest flexural strength, followed by E3, E5 and finally E7. No significant differences (*p* > 0.05) were observed across the E-glass reinforced groups.

### 3.3. Surface Hardness

Figure 3 and Table 4 present surface hardness of the specimens with their mean values and standard deviations. The outcomes of the ANOVA analysis revealed that all reinforced groups exhibited significantly higher Vickers hardness than Group C (*p* < 0.05). However, the findings of the Tukey test indicated that there were no significant differences in surface hardness between the Z1.5 (17.9 ± 0.58 HV_0.30 Kg_) and T1.5 (17 ± 0.63 HV_0.30 Kg_) nanoparticles groups and Group C (17.3 ± 0.49 HV_0.30 Kg_; *p* > 0.05). The highest mean Vickers hardness was found in Group E7 (20.5 ± 1 HV_0.30 Kg_), while the lowest was observed in Group T1.5. When the ratios of filler/fibre were increased, the mean Vickers hardness values positivity increased in ZrO_2_ (R^2^ = 0.9854), TiO_2_ (R^2^ = 0.9917) and E-glass (R^2^ = 0.8674) reinforced groups.

Within the groups reinforced with ZrO_2_ nanoparticles, the Vickers hardness of the specimens in groups Z3, Z5, and Z7 was 18.7 ± 0.55, 19.1 ± 0.58 and 19.6 ± 0.7 HV_0.30 Kg_ respectively, which were significantly higher than that of Group Z1.5. Furthermore, the hardness of the specimens in Group Z7 was statistically higher than those in Group Z3. However, no significant difference in hardness (*p* > 0.05) was found between Groups Z5/Z3 or Z5/Z7 in terms of hardness.

Of the groups reinforced with TiO_2_ nanoparticles, the Vickers hardness in Group T1.5 decreased significantly than that of the TiO_2_ subgroups. No statistically significant difference in hardness was observed between the T3/T5 or T3/T7 groups or group T5/T7.

With regard to the groups reinforced with E-glass fibre, Group E7 demonstrated a significantly (*p* < 0.05) higher Vickers hardness than Groups E1.5 and E3. A further significant difference (*p* < 0.05) was observed between the hardness of the specimens in Group E5 and those in Group E1.5. There were no significant differences (*p* > 0.05) in mean Vickers hardness between Groups E1.5/E3, E3/E5, or E5/E7.

### 3.4. Microstructural Characteristics

The fractured surface of pure PMMA specimens, Groups C displayed a ductile type failure behaviour with irregular areas and small nanopores as shown in Figure 4A. The nanocomposite fractured surface showed signs of particle clustering with small voids (Figure 4B, Z7) and (Figure 4C, T7). This indicated that the distribution of the nanoparticles was not uniform particularly at high particle concentration (i.e., 7 wt.%). Glass fibres are seen embedded within the PMMA matrix either as a single fibre or in bundles Figure 4D, E7). Brittle fracture of glass fibres with small gap between the fibre and matrix were clearly visible.

## 4. Discussions

The null hypothesis was rejected on the basis that variations on both the flexural strength and the surface hardness of the PMMA resins were observed after they were filled with zirconium oxide, titanium oxide nanoparticles, and E-glass fibre.

Various factors were of significance during the process of incorporating filler into PMMA: The distribution of the polymer matrix, the shape and size of the filler, and the level of bonding between the filler and the matrix [10,26,41]. The filler needs to be small enough to produce a homogenous mixtures and can penetrate between the linear macromolecule chains, thereby restricting their movement [9,42]. Previous studies have concluded that the percentage filler should be sufficiently low to ensure the fillers are embedded in the resin [15,18]. In this study, we used sizes of approximately 50 nm ZrO_2_, 25 nm TiO_2_, and 15 μm E-glass fibres that were combined with 90 μm acrylic powder. This prevented a heterogeneous mixture from forming and allowed the nanoparticles to fill any pores that had formed between the polymer particles; as such, the polymer chain movement was inhibited [15].

The flexural strength of dentures plays a vital role in understanding how well a resin will perform under the stress of mastication/chewing [30,35]. Previous studies have found that variations in the ratio of nanoparticles/fibres can have positive or negative impacts on the surface hardness and flexural strength of acrylic resins [10,12,15,19,25]. In the current study we found that the incorporation of ZrO_2_ nanofillers or E-glass fibre within the PMMA matrix enhanced the flexural strength of PMMA in comparison to the specimens in the control group. However, the addition of TiO_2_ nanoparticles to PMMA did not lead to any improvement in flexural strength. As can be observed in Table 4, the flexural strength values of the PMMA increased by 10.3% (*p* < 0.05) when 3 wt.% ZrO_2_ was added to the PMMA resin but it slightly decreased when 7 wt.% ZrO_2_ was incorporated (1%). The addition of 1.5 wt.% and 5 wt.% ZrO_2_ into PMMA also increased the flexural strength of the material; however, the difference was not statistically significant. Thus our proposal was that homogenized blending of fillers into a PMMA matrix might improve the strength of the material [11,42]. However, we observed that the higher ratios of particles led to appearances of clustering, which form spaces. The spaces could explain the decreased strength of the material and the non-homogeneous mixing [7,12,37]. This is supported by the SEM images presented in Figure 4B.

The outcomes of the current study are consistent with the results of previous studies that assessed the impact nano-ZrO_2_ has on the flexural strength of PMMA dentures [18,43]. Nejatian et al. [12] proved that the inclusion of silanated ZrO_2_ into heat-cured PMMA at 10 wt.% reduced the flexural strength of PMMA denture. According to Ergun et al. [19] the addition of various ratios of nano-ZrO_2_ to heat-cured PMMA (5 wt.% 10 wt.%, and 20 wt.%) reduced the flexural strength of the material in comparison to the pure PMMA control group.

Nazirkar et al. [20] found that the flexural strength of the material decreased as the concentration of TiO_2_ increased. Karci et al. [15] also demonstrated that the flexural strength of three different polymerization systems of PMMA acrylic resin fell when a higher concentration of TiO_2_ nanoparticles was included in the resin (3–5 wt.%). In contrast, Naji et al. [44] showed that the flexural strength of conventional PMMA modified with TiO_2_ nanotubes at concentrations of 2.5 wt.% and 5 wt.% were significantly higher than those of the control PMMA. In the current study we found that there was a negative correlation between an increase in TiO_2_ nanoparticle concentration and the flexural strength values of PMMA/TiO_2_ composites. The reduction in flexural strength was observed when the TiO_2_ particles concentration exceeded 1.5 wt.%. The SEM image (Figure 4C) revealed the presence of large porous structures, which could be attributed to agglomeration particles and the non-homogeneous distribution of particles in the PMMA matrix, leading to weak points in the structure [12,21,45]. Previous studies have reported that any hollow space in between the matrix discontinues the stress distribution and may contribute to the inferior mechanical performance of the TiO_2_ particle-reinforced PMMA [12,46].

The research by Lee et al. [25] revealed that the flexural strength of the PMMA significantly increased when the concentrations of silane-treated glass fibres were maintained 6 wt.% and 9 wt.%. Yu et al. [10] also concluded that increasing the concentration of glass fibres significantly increased the flexural strength of the base resin. These findings are in agreement with the results obtained in the present study, which showed that the highest flexural strength values of PMMA/E-glass composites were at the 7wt.% ratio. The PMMA/E-glass matrix interaction is clearly depicted in the SEM image presented in Figure 4D. We observed that the fibres surround the PMMA matrix. This indicated that a good chemical bond had formed between the fibre and the matrix due to the effect the silane treatment had on the fibre [37,41]. This could explain the superior fracture resistance the E-glass fibre reinforced group exhibited in comparison to the other tested polymer composites. Furthermore, due to the level of effective bonding between the fibre and the matrix, upon the complex flexural loading, the fibres might have acted as stress-bearing areas, and slow equal stress distribution may have generated an improved flexural strength and resistance to fibre fracture [10,37].

The Vickers microhardness values in Table 2 indicated that the PMMA reinforced with the three filler materials had a higher surface hardness than the specimens in the control group, with exception of the groups Z1.5 and T1.5. The specimens in Group T1.5 exhibited an insignificantly lower surface hardness than those in the control group. At the highest concentration, the E-glass fillers had increased the hardness of the composites by the highest amount followed by the ZrO_2_ and then the TiO_2_. This result is in agreement with previous research, which has found that the addition of glass fibre [9,36], ZrO_2_ [5,18] and TiO_2_ [12,15] nanoparticles in PMMA at different concentrations statistically improved the surface hardness of PMMA denture bases. The increase in filler content enhanced the surface hardness up to the point at which the optimal level was achieved [22]. This can be attributed to the fact that, when the optimal level of filler for the matrix is reached through magnetic stirrers and speed mixers, the agglomeration of the composite is reduced [5,22].

Generally, the most viable explanation for the increase in flexural strength and surface hardness of PMMA/filler observed in the current study was the use of the 3wt.% silane coupling agent, which helped to improve the chemical bond between the filler and the PMMA matrix [9,18]. This entails that a greater amount of energy is required to break the chemical bonds that form between the materials [5,30]. A further explanation for this improvement was the homogeneous filler distribution that was achieved by using an ultrasonic speed mixer to combine the filler particles in the monomer. However, the decreased flexural strength observed at the high percentage of nanoparticles could be attributed to the higher levels of filler in the mix [12,22]. One explanation for this could be the fact that flexural strength is a bulk property in comparison to surface hardness. Once the saturation point has been reached, the resin cannot absorb any further filler. Adding more filler after the resin reaches the saturation point disturbs the continuity of the resin matrix and, subsequently, weakens the properties of the composite materials [5,13].

## 5. Conclusions

The optimal filler concentrations for reinforcing PMMA denture base resins from the flexural strength perspective were 3–5 wt.% ZrO_2_, 1.5 wt.% TiO_2_, and 3–7% wt.% E-glass fibre. This study found that the surface hardness of the reinforced PMMA generally increased as the content of the fillers increased. A significant increase in the hardness of the composites was observed for all the fillers at all concentrations with exception of the 1.5 wt.% concentration of TiO_2_ and ZrO_2_ nanoparticles. Therefore, a filler concentration greater than 3 wt.% could be considered suitable for producing composites that offer a significantly higher hardness. The findings of this study indicated that the addition of E-glass fibre to PMMA delivered the greatest improvement in terms of mechanical properties, followed by ZrO_2_ and then TiO_2_. Thus, improving the mechanical properties of PMMA denture bases through the use of the fibre and filler can produce dentures that can achieve longer clinical service.

## Figures and Tables

**Figure 1 materials-14-02659-f001:**
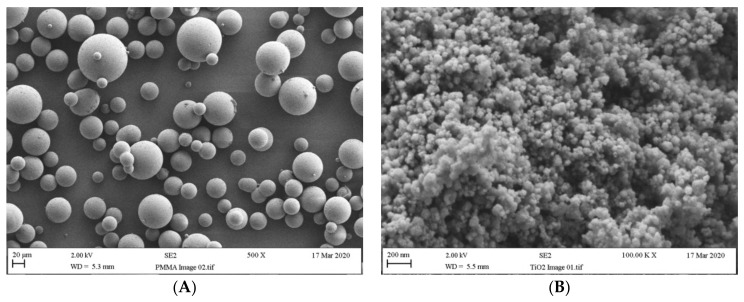
Particle/fibre size of (**A**) PMMA powder, (**B**) ZrO_2_ nanoparticles, (**C**) TiO_2_ nanoparticles and (**D**) E-glass fibre.

**Figure 2 materials-14-02659-f002:**
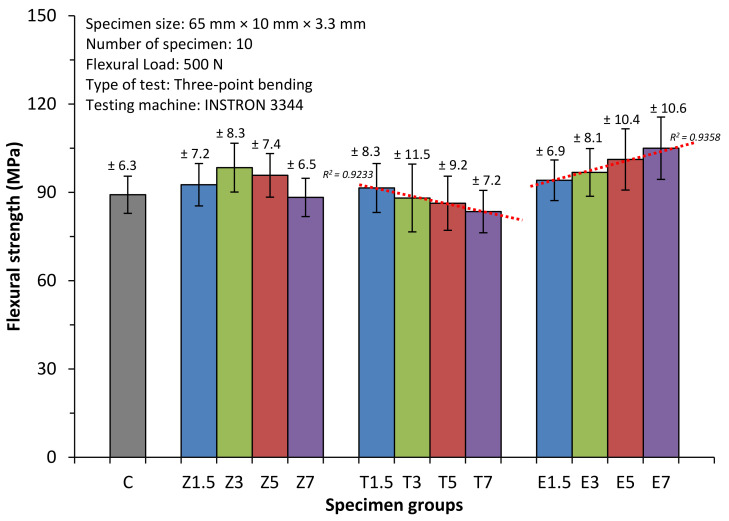
Mean flexural strength of all specimen groups with standard deviation values.

**Figure 3 materials-14-02659-f003:**
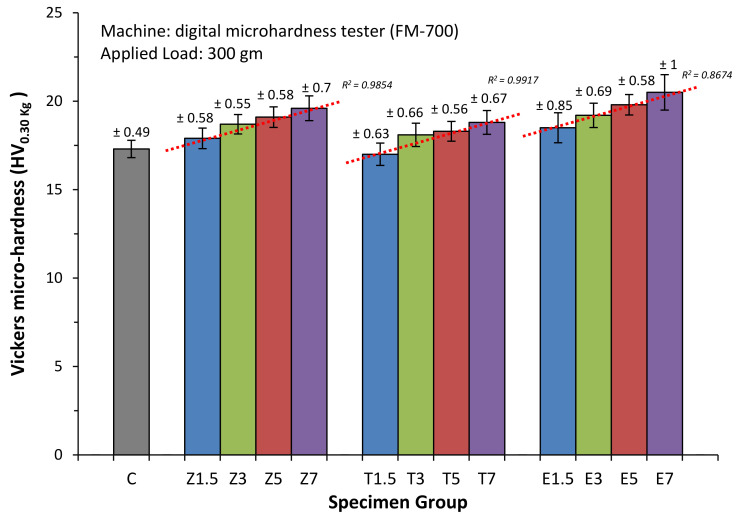
Mean surface hardness of all specimen groups with standard deviation values.

**Figure 4 materials-14-02659-f004:**
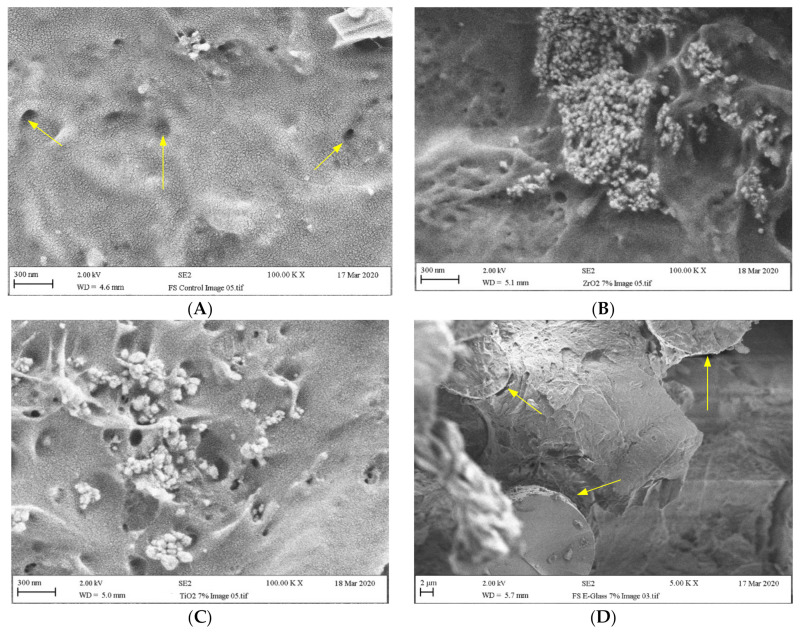
Fractured surfaces of (**A**) pure heat-cured PMMA (arrow showing pores) and reinforced composites (Group C) with (**B**) ZrO_2_ nanoparticle (Z7), (**C**) TiO_2_ nanoparticle (T7), and (**D**) E-glass fibre (E7; arrow showing gap between fibre and matrix).

**Table 1 materials-14-02659-t001:** Materials used in the experiments.

Material	Composition and Specifications	Manufacturer
Lucitone-199^TM^	Heat-polymerized acrylic resin Powder: PMMA; Monomer: MMA	Dentsply International, York, PA, USA
Zirconium oxide	Zirconium(IV) oxide-yttria stabilized, nanopowder, <100 nm particle size	Sigma Aldrich, Gillingham, UK
Titanium oxide	Titanium(IV) oxide, anatase, nanopowder, <25 nm particle size	Sigma Aldrich, Gillingham, UK
E-glass fibre	3 mm in length, 15 μm in diameter	Hebei Yuniu Fibreglass, Xingtai, China
Ethanol	Ethanol, absolute (C_2_H_6_O, EtOH)	Fisher Scientific, Loughborough, UK
Silane coupling agent	3-(Trimethoxysilyl)propyl methacrylate, assay 98%	Sigma Aldrich, Gillingham, UK

**Table 2 materials-14-02659-t002:** Specimen grouping and coding.

Materials Group	Group/Subgroup Code	Material Description	Number of Specimens
Control	C	PMMA acrylic resin	10
ZrO_2_ nanoparticle	Z1	PMMA acrylic resin + 1.5 wt.% ZrO_2_	10
	Z3	PMMA acrylic resin + 3 wt.% ZrO_2_	10
	Z5	PMMA acrylic resin + 5 wt.% ZrO_2_	10
	Z7	PMMA acrylic resin + 7 wt.% ZrO_2_	10
TiO_2_ nanoparticle	T1	PMMA acrylic resin + 1.5 wt.% TiO_2_	10
	T3	PMMA acrylic resin + 3 wt.% TiO_2_	10
	T5	PMMA acrylic resin + 5 wt.% TiO_2_	10
	T7	PMMA acrylic resin + 7 wt.% TiO_2_	10
E-glass fibre	E1	PMMA acrylic resin + 1.5 wt.% E-glass	10
	E3	PMMA acrylic resin + 3 wt.% E-glass	10
	E5	PMMA acrylic resin + 5 wt.% E-glass	10
	E7	PMMA acrylic resin + 7 wt.% E-glass	10

**Table 3 materials-14-02659-t003:** Quantities of acrylic resin powder, monomer, and filler used in each group.

Filler Concentration	Filler in Each Mould (g)	PMMA Powder (g)	MMA Monomer (mL)
0% (Control)	0.0	12.00	5.70
1.5%	0.18	11.82	5.70
3%	0.36	11.64	5.70
5%	0.60	11.40	5.70
7%	0.84	11.16	5.70

**Table 4 materials-14-02659-t004:** Mean and standard deviation (SD) of flexural strength and surface hardness values for the tested groups.

Group		Flexural Strength (MPa) Mean ± SD	Surface Hardness (HV_0.30 Kg_) Mean ± SD
Control	C	89.2 (6.3) ^ACD^ *	17.3 (0.49) ^AD^
ZrO_2_	Z1.5	92.6 (7.2) ^AB^	17.9 (0.58) ^A^
	Z3	98.4 (8.3) ^B^	18.7 (0.55) ^B^
	Z5	95.8 (7.4) ^AB^	19.1 (0.58) ^BC^
	Z7	88.3 (6.5) ^A^	19.6 (0.70) ^C^
TiO_2_	T1.5	91.5 (8.3) ^C^	17.0 (0.63) ^DE^
	T3	88.1 (11.5) ^C^	18.1 (0.66) ^E^
	T5	86.3 (9.2) ^C^	18.3 (0.56) ^E^
	T7	83.5 (7.2) ^C^	18.8 (0.67) ^E^
E-glass fibre	E1.5	94.1 (6.9) ^DE^	18.5 (0.85) ^F^
	E3	96.8 (8.1) ^DE^	19.2 (0.69) ^FG^
	E5	101.2 (10.4) ^E^	19.8 (0.58) ^GH^
	E7	105 (10.6) ^E^	20.5 (1.0) ^H^

* Similar superscript letters in the same column indicate no significant difference between each of the reinforced groups and the PMMA acrylic resin control group (*p* > 0.05).

## Data Availability

The data presented in this study are available within the article.

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
