# Peer review of "Flexural Strength and Hardness of Filler-Reinforced PMMA Targeted for Denture Base Application"

_materials, 2021, doi:10.3390/ma14102659_

Round 1

Reviewer 1 Report

Congratulations!

The material is presented very clearly and the obtained data could have a practical impact on improving dental prosthesis technology and patient satisfaction. I encourage you to try to use these results in order to develop new formulas for PMMA. 

Author Response

Kindly see the attachment.

Reviewer 2 Report

Title: Flexural Strength and Hardness of Filler-Reinforced PMMA Targeted for Denture Base Application

In this work the Authors studied the effects of ZrO2 nanoparticles, TiO2 nanoparticles, and E-glass fibers added as fillers to heat-cured PMMA samples in terms of mechanical properties.

The Abstract is excessively detailed and reports unnecessary information, such as the large part of the results obtained (e.g. the trend observed for flexural strength and surface hardness) which are discussed later in the manuscript and can be briefly reported, in case, in the conclusions. The Authors should consider modifying the abstract according to the Guidelines for Authors in which is recommended 200 words as maximum length.

Line 261: typing error

The conclusions lack of clarity. The purpose of the work should be briefly summarized. Also, the Authors should consider underlining the most significant results obtained (e.g. the extent of improvement achieved) which can support the discussion.

Reviewer 3 Report

With independence of the material’s application (denture’s plate), the paper talks about the flexural strength and the hardness of PMMA with different reinforcements (being these differences focused the type of the reinforcement and the volume fraction of that). To obtain these two mechanical properties the authors make a three-point bending test on four samples with ten specimens in each one. I wish to write the following comments to authors, I hope that my comment will be helpful.

  1. In my opinion this work is interesting but, being honest, I think that is very difficult to read because is too long. In my opinion to use 16 pages to write this work is excessive. A study like this should be solved with a lower number of pages. The fourth point of the work, the discussion’s point, can be considered as an example. This point is very long and is written like the introduction’s point of the work. 29 of 47 references are cited in this fourth point of the work, using the following expressions: they studied, they found, they concluded that, they also found, they showed that, they stated that, … In my opinion the discussion’s point should be centered more in “we” (we obtain, we found, we observed that, our proposal is that, …) than in “they”. There are too many citations of external works and results in this point, I think that this point should be focused on the internal work of the authors (because the main objective of this work is not a compilatory study of external works). Well, like I wrote before, a large part of the discussion’s point is written in the same form that the introduction’s point (the state of the art). This point is too big, and I believe that should be reduced.
  2. The numbering of the figures is not well cited in the text. The letter appears but not the corresponding number.
  3. 4(a): is difficult to see the details clearly.
  4. 4: in lines 325-326 authors write about cracks. Where are they?
  5. In the point 3.1 the authors show different values of particle size. How do the authors measure that? This question is focused, specially, for the case of Fig 1 (B) and (C). To obtain the different measures from these two cases can’t be easy!

Kind regards.
